# Thyroid Hormone and Heart Failure: Charting Known Pathways for Cardiac Repair/Regeneration

**DOI:** 10.3390/biomedicines11030975

**Published:** 2023-03-21

**Authors:** Polyxeni Mantzouratou, Eleftheria Malaxianaki, Domenico Cerullo, Angelo Michele Lavecchia, Constantinos Pantos, Christodoulos Xinaris, Iordanis Mourouzis

**Affiliations:** 1Department of Pharmacology, University of Athens, 11527 Athens, Greece; 2Centro Anna Maria Astori, Istituto di Ricerche Farmacologiche Mario Negri IRCCS, 24126 Bergamo, Italy

**Keywords:** thyroid hormone, thyroid receptors, low T3 syndrome, heart failure, coronary disease, cardiac remodeling

## Abstract

Heart failure affects more than 64 million people worldwide, having a serious impact on their survival and quality of life. Exploring its pathophysiology and molecular bases is an urgent need in order to develop new therapeutic approaches. Thyroid hormone signaling, evolutionarily conserved, controls fundamental biological processes and has a crucial role in development and metabolism. Its active form is L-triiodothyronine, which not only regulates important gene expression by binding to its nuclear receptors, but also has nongenomic actions, controlling crucial intracellular signalings. Stressful stimuli, such as acute myocardial infarction, lead to changes in thyroid hormone signaling, and especially in the relation of the thyroid hormone and its nuclear receptor, which are associated with the reactivation of fetal development programmes, with structural remodeling and phenotypical changes in the cardiomyocytes. The recapitulation of fetal-like features of the signaling may be partially an incomplete effort of the myocardium to recapitulate its developmental program and enable cardiomyocytes to proliferate and finally to regenerate. In this review, we will discuss the experimental and clinical evidence about the role of the thyroid hormone in the recovery of the myocardium in the setting of heart failure with reduced and preserved ejection fraction and its future therapeutic implications.

## 1. Introduction

Heart failure (HF) is a clinical syndrome with specific symptoms and signs (e.g., breathlessness, peripheral edema) due to structural or functional cardiac abnormalities that result in elevated intracardial pressure and/or impaired cardiac output [1]. Although the prognosis of patients has considerably improved over the last few decades, HF still affects more than 64 million people worldwide, having a serious impact on their survival and quality of life [2]. Hospitalizations because of HF represent 1–2% of all admissions in Western countries, and HF is the most common cause of hospitalization of individuals >65 years. Around 30–40% of HF patients have a history of hospital admission, and 50% are readmitted within 1 year of their initial diagnosis. Mortality is high, with the 1-year risk being between 15–30% and the 5-year risk up to 75% in several populations [2]. The current health care costs per year for every HF patient are up to EUR 25,000 in the Western world, while the increasing prevalence of HF is expected to lead to prohibitive costs, even for developed countries [2]. Therefore, there is an urgent need to better understand the pathophysiology and molecular basis of HF in order to develop new therapeutic approaches.

Thyroid hormone (TH) signaling, evolutionarily conserved, controls fundamental biological processes and has a crucial role in development and metabolism [3]. The active form of TH is L-triiodothyronine (T3), which regulates important gene expression by binding to its nuclear receptors, TH receptors (TRs; TRalpha1-2 and TRbeta1-2) [4]. In the heart, ventricular cardiomyocytes mostly express TRalpha1, while both TRalpha1 and TRbeta1 are expressed in the peripheral ventricular conduction system [5]. TRs are transcription factors that reside in the nucleus but also rapidly shuttle between the cytoplasm and nucleus. Genomic actions of TRs include interactions with TH response elements in specific genes [6]. Nongenomic actions of TH regulate important intracellular signaling pathways and are mediated via cytosolic TRs or via the membrane integrin receptor αvβ3 [6].

One of the best-characterized TRs in mammals is TRalpha1, which is present and well described in the heart [7]. Interestingly, during fetal life, due to low levels of T3, TRalpha1, highly expressed, is in an unliganded state, acting as an aporeceptor, repressing the expression of the adult gene program and allowing the proliferation of cardiomyocytes and an increase in cardiac mass. After birth, a burst of T3 results in the liganded state of TRalpha1, which, acting as a holoreceptor, triggers the expression of adult genes promoting heart maturation and development [8]. It is of great interest that TRalpha1 seems to act like a “molecular switch” during heart development, since its status as an apo- or a holoreceptor controls the proliferation and differentiation of the cardiomyocytes, and thereby this role can be of great clinical relevance [9]. Indeed, in a recent study, a dominant negative TRalpha1—which is unable to bind T3 and become a holoreceptor—in a mouse heart, prevented myocardial cells from complete differentiation and permitted their proliferation—and thus the regeneration—of the myocardium after acute myocardial infarction (AMI) [10]. Importantly, in adult life after stressful stimuli and during disease states, such as in HF, the fetal profile of the TRalpha1-T3 axis recurs. This recapitulation of fetal-like features of the signaling may be an incomplete effort of the myocardium to recapitulate its developmental program and enable cardiomyocytes to proliferate and finally to regenerate [8]. In organs with very limited regenerative potential, such as the heart, the fetal profile of the TRalpha1-T3 axis during stress results in cell hypertrophy without progressing to reductive mitosis. This hypertrophy can temporally compensate the dysfunction, but when accompanied by adoption of a low energy profile, it becomes maladaptive and leads to failure [11]. Therefore, after stress, a drop in T3 serum levels takes place—a phenomenon known as “low T3 syndrome”—whilst the unliganded TRalpha1 moves to the nucleus to induce dedifferentiation and reactivation of fetal genes, as well as growth in cardiomyocytes. The main regulators of T3 concentration at the tissue level are enzymes called deiodinases, which mediate the activation or inactivation of TH. Types 1, 2, and 3 iodothyronine deiodinases have unique catalytic properties and tissue distributions [12,13,14]. Deiodinase type 3 (D3) has a high affinity in inactivating T3, playing a critical role in T3 availability in the systematic level as well as locally in the injured tissue [13]. Interestingly, D3 is highly expressed in the heart after myocardial injury [15]. In addition, the shuttling of TRalpha1 to the nucleus is shown to be controlled by the activation of the adrenergic system [16], which is thought to be involved in fetal gene reactivation and cardiac remodeling. Milestones of the fetal-like shift in the myocardium are an increase in the beta myosin heavy chain (the predominantly fetal type of myosin, which is slower and less energy-consuming than the adult alpha type of myosin), the decreased ratio of sarcoplasmic reticulum-calcium ATPase (SERCA) to phospholamban (PLN), and the “energy shift” of the stressed myocardium to the use of glucose as an energy substrate instead of fatty acids (like in fetal life) [17,18]. Although this transcriptional shift in the short term allows the cardiomyocytes to recover to a less energy-demanding state by having a lower maximum shortening velocity [19], in the long term it becomes detrimental, leading to further deterioration of the organ’s structure and function [20] (Figure 1).

Exogenous administration of T3 in the setting of acute or chronic heart injury turns TRalpha1 to its holoreceptor form, promoting the expression of the adult transcriptional program and ameliorating cardiac structure and function [9]. In accordance, a series of epidemiological studies reveal an association of low T3 levels after stress with adverse clinical outcomes in cardiological patients such as the ones with HF, while exogenous T3 treatment has promising results [7].

In this review, we will discuss the experimental and clinical evidence about the role of TH in the recovery of the myocardium in HF with reduced and preserved ejection fraction (HFrEF, HFpEF) and its future therapeutic implications.

## 2. TH and HFrEF

Cardiac dysfunction results from myocardial injury and/or changes in the viable nonischemic myocardium, a process known as cardiac remodeling. This response is characterized by altered cardiac chamber geometry, a shift in cardiomyocyte protein expression to a fetal pattern, energy deficit, and the induction of fibrosis [21,22]. Despite the advances in current treatments, cardiac remodeling occurs in nearly 30–40% of patients with AMI treated with reperfusion, and results in progressive dilatation and HF, making coronary artery disease the most common cause of HfrEF [23].

### 2.1. Preclinical Studies

Significant changes in TH signaling take place during cardiac remodeling after acute myocardial injury. In a model of AMI in rats, T3 levels drop within a week, and TH-related genes, such as myosin isoforms and SERCA, normalize after TH administration [18]. A distinct pattern of TRalpha1 alteration is also present after acute injury and in the course of HF. During the compensatory stage of HF, TRalpha1 expression increases and declines thereafter with the progression to the noncompensatory left ventricular dysfunction [24]. Interestingly, the pharmacological inactivation of TRalpha1 in the postischemic myocardium in mice results in a decreased ratio of SERCA to PLN and in activated proapoptotic p38 mitogen-activated protein kinase (MAPK). Therefore, there is a further dramatic deterioration of postischemic heart function [25], and this makes the emerging role of TH signaling in the response to stress after myocardial injury clear. In more detail, mice with AMI, which are treated with the selective TRalpha1 inhibitor debutyl-dronedarone, exhibit a significantly depressed left ventricular (LV) function with lower ejection fraction (EF) and higher wall tension index compared to the control group. These changes are accompanied by a marked activation of the proapoptotic p38 MAPK, known for its negative inotropic effect. On the other hand, when TH is administered to the same model of acute myocardial injury and HFrEF, in a replacement dose, there is a significant improvement of LV function and geometry, a decrease in the expression of the fetal type of myosin (beta myosin), and an enhancement of the prosurvival signaling Akt [26]. Likewise, in a rat model of ischemic heart disease, TH treatment results in improved LV geometry as well as function, inhibits the expansion of the scar over time [27], and reduces apoptosis by activating Akt again [28]. Early and constant replacement of T3 levels in the same model also prevents the progression towards HF, likely due to increased capillary formation and mitochondrial protection [29]. Even with the presence of comorbidities, such as diabetes, TH administration improves wall stress, increases cardiac mass, and ameliorates cardiac remodeling [30]. Moreover, TH pretreatment in ex vivo rat models of ischemia show a protective effect against ischemia–reperfusion injury by suppressing the activation of the proapoptotic p38 kinase cascade [31], while T3 treatment at reperfusion significantly helps the recovery of function and reduces apoptosis and tissue necrosis [32].

### 2.2. Epidemiological Studies

On the basis of the aforementioned experimental evidence, TH signaling appears to be essential for the response of the myocardium to stress in the setting of acute myocardial injury. In fact, in numerous epidemiological studies, researchers have noticed the association of TH levels with clinical outcomes in the clinical conditions of acute myocardial ischemia and HF. In patients with AMI followed by percutaneous coronary intervention (PCI), the levels of T3 six months after the incidence appear to be an independent predictor of recovery of cardiac function [33]. In the same setting, patients younger than 75 years old with low free T3 levels have higher mortality [34], while patients with elective or primary PCI and subclinical hypothyroidism have worse outcomes of repeat revascularization and cardiac death following PCI [35]. Interestingly, patients with ST segment elevation myocardial infarction, treated with PCI who developed cardiogenic shock, have lower free T3 (FT3) and higher free T4 (FT4) upon admission compared to those without cardiogenic shock. During 2.5 years of follow-up, those with low FT3 (<2.85 pg/mL) and high FT4 (≥0.88 ng/dL) have the highest all-cause mortality (18.2%), while those with high FT3 and low FT4 have the lowest (3.8%) [36]. The prognostic value of the ratio FT3 to FT4 is also examined in patients with myocardial infarction with nonobstructive coronary arteries. Patients with lower ratio have a higher incidence of major adverse cardiovascular events (MACE) (10.0%, 13.9%, 18.2%; *p* = 0.005) over the median follow-up over 41.7 months. The risk of MACE also increases when FT3/FT4 is decreased after multivariate adjustment, and a low level of FT3/FT4 ratio is strongly connected to a poor prognosis [37]. The association between low T3 levels and mortality was once again highlighted by a ThyAMI-1 study in which patients with AMI and low T3 syndrome had significantly higher all-cause mortality [38]. In patients with chronic HF, Pingitore et al. also showed that low T3 levels are an independent predictor of mortality [39]. Significantly worse one-year all-cause mortality was also identified in patients with decompensated acute HF [40]. In those patients, levels of T3 are proposed to be used for their risk stratification [41]. In hospitalized patients with chronic HF, low T3 is associated with higher cardiac and all-cause mortality [42], while exercise capacity, as an indicator of functional status, is reduced in patients with HF and low T3 levels [43]. Kannan et al. showed that in patients with pre-existing HF, subclinical hypothyroidism with thyroid stimulating hormone (TSH) ≥7 mIU/L and low T3 levels is associated with poor prognosis [44]. Interestingly, in patients with idiopathic dilated cardiomyopathy, low FT3 level is also associated with myocardial fibrosis and perfusion/metabolism abnormalities as evaluated by cardiac magnetic resonance imaging (CMR), single-photon emission computed tomography (SPECT), and positron emission tomography (PET) [45].

### 2.3. Clinical Trials

Based on the above clinical findings and on the aforementioned experimental evidence, TH treatment has been used in several trials in patients with HF or ischemic heart disease. A study of acute intravenous administration of T3 in a small number of patients with advanced HF established the basis for further investigation into the safety and potential benefits of TH treatment. An intravenous bolus dose of T3 was administered to 23 patients with advanced HFrEF without adverse events. T3 administration increased cardiac output along with a reduction in systemic vascular resistance in patients receiving the largest dose [46]. In another study, short-term synthetic L-T3 replacement therapy in patients with HF had positive results [47]. A total of 20 patients with ischemic or nonischemic HF and low T3 syndrome, clinically stable, were enrolled. The study group (10 patients) underwent 3-d synthetic L-T3 infusion in order to restore normal T3 levels as rapidly as possible without adverse effects. The control group (10 patients) underwent placebo infusion. The main finding of the study was the positive effect in the cardiac function and neuroendocrine profile of the patients. Thus, the study group had an improved stroke volume of the LV and a significantly reduced noradrenalin, aldosterone, and N-terminal-pro-B-type natriuretic peptide (NTproBNP) [47]. On the contrary, in a small randomized, double-blind, crossover, placebo-controlled interventional study, oral T3 given twice daily for three months in patients with low T3 levels, chronic HF, and modestly reduced LVEF did not improve cardiac function and neurohormonal profile [48]. Although this could be partly explained by the small sample size and the oral type of treatment for a relatively short time of period, another randomized double-blind, placebo-controlled trial had different results. In more detail, fifty adult patients with clinically stable HF and low T3 levels received oral liothyronine or placebo for 6 weeks. Liothyronine had an important impact in patients’ functional status by significantly improving the 6 min walk test, their neurohormonal profile, and cardiac LVEF [49]. Although the above studies examine the effect of T3 in patients with HF and low T3 syndrome, an ongoing multicenter, open-label, randomized, parallel group trial (ThyroHeart-CHF), aims to evaluate the efficacy and safety of levothyroxine replacement on the exercise capability in chronic systolic HF patients with subclinical hypothyroidism without examining T3 levels [50]. In more detail, eligible patients have to be 18 years or older, with systolic HF (New York heart association (NYHA) class II–III), LVEF ≤ 40%, and subclinical hypothyroidism. They will be randomly 1:1 assigned in order to receive thyroxine replacement therapy along with standard chronic HF treatment or only standard HF therapy. The initial levothyroxine dose will be 12.5 μg once a day and will be titrated until TSH normalization (T3 or T4 values are not examined). The primary endpoints include the difference in 6 min walk test results between 24 weeks and baseline. Secondary endpoints are the differences in neurohormonal and serum lipid profiles, changes in the NYHA classification, cardiovascular death, rehospitalization, differences in heart structure and function as assessed by echocardiogram and CMR imaging measures, and Minnesota living with heart failure questionnaire results between 24 weeks and baseline.

In the last few years, several phase II studies have been conducted for the first time to investigate the potential of TH treatment after AMI to prevent the development of cardiac remodeling and HF. Pingitore et al. investigated whether TH replacement therapy is safe in patients after AMI with low T3 syndrome and whether it has an impact on the infarct size and LV volumes and function. In this study, 37 patients with AMI were randomly treated or untreated with T3 for 6 months in addition to the standard therapy. At discharge and at 6 months, the LV volumes, LVEF, wall motion score index (WMSI), and infarct extent were measured by CMR. At follow-up, there was a significant reduction in WMSI for patients in both groups, while the difference value (discharge/follow-up) was significantly higher in the T3-treated group. In addition, stroke volume increased significantly at follow-up after T3 treatment, which appeared to be safe and able to improve regional dysfunction in patients with AMI. However, no effect of TH replacement therapy was found in infarct extent, LV volumes, and EF [51].

In another double-blind, randomized clinical trial, 95 patients with AMI were randomized to receive either levothyroxine (starting at 25 µg) or a placebo for 52 weeks, and cardiac function was assessed by CMR imaging at baseline and at the end of the study. Treatment with levothyroxine was shown to be safe but did not significantly improve EF, LV volumes, or infarct size after 52 weeks [52]. The dose and timing of administration may play a significant role in these trials. Upon ischemic stress, there is an impaired conversion of T4 to T3, while T3 inactivation is increased due to alterations in deiodinase activity. Furthermore, changes at the level of TH receptors take place and modify the response of the myocardium to THs [7]. More recently, a pilot, randomized, double-blind, placebo-controlled trial (ThyRepair study) investigated potential effects of acute, high-dose LT3 treatment in patients with ST-elevation anterior AMI. LT3 treatment started after primary PCI with an intravenous bolus injection of LT3 followed by a constant infusion for 48 h. Data were analyzed from 37 patients who had CMR at hospital discharge and 6 months follow-up. Acute LT3 treatment resulted in significant lower LV end-diastolic volume index and LV systolic volume index at hospital discharge, while CMR infarct volume was lower in the LT3-treated group at 6 months. These findings may be of high clinical relevance. Early LV dilatation, as assessed by CMR, was shown to carry 57% long-term mortality vs. 27% and 26% of late and absence of dilatation, respectively [53]. Moreover, after adjustment for LVEF and age, early dilatation was the exclusive independent predictor of long-term mortality [53]. The primary endpoint of the present study, LVEF%, was found increased in the LT3-treated group of compared to the placebo, although without statistical significance. It seems that at this stage, LV with higher dilatation, which was found in untreated patients, could contribute to higher LVEF% via Starling’s law effect [53]. However, LVEF% difference between groups, evident at 6 months of follow-up, was at a magnitude of 5 units, and this is of clinical relevance since an LVEF% change of higher than 5 units is a powerful predictor of both HF hospitalizations and survival [54], although a bigger sample size was necessary to designate a significant change in LVEF% at this stage. Furthermore, it is also interesting that ECG QRS duration at 6 months follow-up was significantly lower in the LT3-treated group, indicating a potential positive effect of T3 on electrical remodeling of the heart. In fact, the prolonged duration of QRS after AMI shows adverse electrical remodeling and correlates with increased mortality [55]. Despite a tendency for increased incidence of atrial fibrillation during the first 48 h, serious, life-threatening events related to LT3 treatment were not observed [56].

Interestingly, TH therapy has been also used for the hemodynamic support of heart donors in cardiac transplantation, having a protective role against ischemic injury. This effect was evident in a series of 66,629 organ donors, where T3 or T4 treatment was connected to the attainment of a significantly higher number of cardiac grafts. Astonishingly, this effect was also associated with increased graft survival after transplantation and was also independent of other factors [57,58].

We could consider several variables in any attempt to explain inconsistencies between the above trials, such as the time and dosing of administration, intrinsic methodological differences in the analysis tools, and clinical endpoints and the nature and severity of injury/disease. Based on the above, the exogenous administration of TH seems to be both safe and beneficial for patients with HF and ischemic heart disease. However, large-scale trials are needed in order to validate these results (Table 1).

### 2.4. TH and HFpEF

Although the prevalence of HFpEF is rapidly growing, due to the aging population and an increase in pathological conditions such as hypertension and diabetes [59], its therapy remains challenging [60]. Pathophysiologicaly, diastolic impairment with abnormal relaxation and increased passive stiffness predominate [61]. Myocardial stiffening can be attributed to the giant cytoskeletal protein titin as well as to the extracellular matrix, and HFpEF patients have both increased collagen content and titin-dependent stiffness. Changes in calcium homeostasis, including increased diastolic calcium levels [62], are also quite important contributors to abnormal relaxation in HFpEF, while impaired bioenergetics have also been proposed as a key mechanism [63]. Interestingly, TH biological actions can pleiotropically affect the underlining pathophysiology of HFpEF. It is known that THs not only stimulate cell growth and neoangiogenesis, but also decrease cardiac fibrosis by enhancing metalloproteinase activity [64]. It is also of great importance that THs enhance the expression of genes encoding SERCA and negatively regulate the transcription of PLN. The increase in SERCA and the inhibition of PLN not only increase the calcium available in systole, but also improve its reuptake into the sarcoplasmic reticulum during relaxation of the heart [65]. In terms of bioenergetics, importantly, THs stimulate cardiac mitochondrial biogenesis and improve oxidative phosphorylation, and this can have a great impact not only in systolic but also in diastolic cardiac function [66]. 

Indeed, abnormal and especially low TH levels are associated with diastolic cardiac impairment [67]. Patients with subclinical hypothyroidism that were evaluated by Doppler echocardiography showed significant prolongation of the isovolumic relaxation time, reduced E/A ratio, and an increased A wave [68]. In the subgroup of patients that were re-evaluated after TH profile normalization, diastolic abnormalities were reversed [68]. In patients with overt HFpEF, subclinical hypothyroidism and low T3 syndrome are quite common. The inflammatory process in HFpEF, along with the intracellular hypoxia, may contribute to the increased D3 gene expression, which results in the degradation of T3 into inactive metabolites and in local hypothyroidism [15,69]. In an interesting study, among 89 patients with HFpEF, 22% exhibited low T3 levels, which were associated with markers of severity, such as B-type natriuretic peptides as well as echocardiographic parameters of diastolic impairment [70]. Data from animal models of HFpEF suggest an improvement in diastolic function after TH treatment. Longstanding hypertension, in animal models, results in low T3 in serum and heart tissue along with increased collagen and a fetal phenotype shift in myosins. Treatment with low doses of T3 in the long term not only normalizes serum and cardiac tissue T3 levels, but also restores α/β myosin protein levels, collagen, and systolic wall stress, tending to improve diastolic function [71]. In a rat model of type II diabetes, T3 treatment prevents tissue fibrosis, cardiomyocyte dedifferentiation and cytoarchitectural alterations, reverses the diabetes-induced reactivation of fetal genes and pathological growth, and improves myocardium ultrastructure (unpublished data).

There are no clinical trials that have investigated the effects of TH treatment in patients with HFpEF. Interestingly, a phase II randomized trial aims to determine the feasibility, safety, and preliminary efficacy of oral LT3 therapy in patients with HfpEF, and is expected to be completed in 2023. The design includes a treatment during an approximate period of 8 weeks, with every week titration of study drug for 4 weeks, a maintenance dose of 4 weeks, then a 2-week washout, and finally crossing over to the other arm (drug/placebo). LT3 is titrated based on serum T3 levels. The minimum LT3 dose is set to 2.5 mcg three times daily and the maximum LT3 dose to 12.5 mcg three times daily. Endpoints of efficacy include the peak maximal rate of oxygen consumption during exercise (VO2 Max), quality of life, and NT-proBNP levels [72].

Effective treatments for HFpEF are lacking. Thus, understanding the potential therapeutic role of TH in this syndrome could prove the missing link in the quest for novel treatments for diastolic dysfunction.

## 3. The Challenge of Clinical Translation

Exogenous administration of TH in experimental models of HF has showed, as already mentioned, great therapeutic potential. Nevertheless, translating this new therapeutic strategy into clinical practice has proven a great challenge. The coronary drug project was the first randomized, placebo-controlled trial that investigated the effect of a TH analog, dextrothyroxine (DT4), in patients after AMI. In this old study, in an era when reperfusion with PCI or thrombolysis did not exist, DT4 was given at high doses for several months, and resulted in small but significant increases in arrhythmia and mortality, which caused the discontinuation of this treatment arm [73]. In another, more recent study, patients with chronic HF were treated with excessive doses of another TH analog, diiodothyropropionic acid (DITPA), for 6 months. DITPA improved some hemodynamic parameters, such as cardiac index and vascular resistance, but was poorly tolerated, mainly due to fatigue [74]. Several preclinical studies published during the last few years have increased our knowledge about the favorable or detrimental actions of TH treatment in cardiovascular diseases. In the injured or failing heart, the conversion of T4 to T3 is impaired and deactivation of T3 is enhanced due to increased D3 activity. Furthermore, changes occur in the expression and/or shuttling of TRs, resulting in an altered response of the myocardium to THs compared to the normal myocardium [7]. Thus, high doses of TH are needed in order to increase T3 levels locally in the diseased heart. However, in dose-dependent preclinical studies, long-term administration of high doses of T4 and T3 have shown adverse effects in cardiac remodeling and increased mortality [26]. The dose, timing, and duration of administration are probably critical points in translating the beneficial effects of TH. As reported above, three recent phase II randomized, double-blind, placebo-controlled trials were performed in patients with AMI, and three different therapeutic approaches were tested: low-dose (replacement) LT4 treatment for 52 weeks [52]; low-dose (replacement) LT3 treatment for 6 months [51]; and high-dose (7–10 times the replacement dose), acute LT3 treatment for 48 h [56]. The safety of the treatment was observed in each case. LT4 did not show any beneficial effect, while some secondary beneficial effects were seen after LT3 treatment for 6 months. On the other hand, acute, high-dose LT3 showed great therapeutic potential, improving LV dilatation and reducing infarct volume. However, some concerns were raised due to a trend towards a higher incidence of early reversible atrial fibrillation in LT3-treated patients. Furthermore, there was a modest increase in heart rate, and nervousness was observed in some patients during the period of administration [56].

A different or complementary therapeutic approach could include the development of drugs that enhance local tissue T3 levels indirectly by potentiating the conversion of T4 to T3 or by inhibiting D3 activity [75]. In fact, a beneficial role of vitamin D supplementation has been shown in the conversion of LT4 to LT3 via D2 in experimental studies. In addition, in a recent clinical study in AMI patients, a relationship between hypovitaminosis D and LT3 levels has been found, indicating that vitamin D supplementation could potentially act to restore local T3 levels [76].

The development and synthesis of novel TH analogs could also prove valuable in order to selectively target specific TRs [77]. In this regard, TH analogs lacking iodine molecules would be resistant to deiodinase activity and could specifically activate TR receptors of interest in low doses.

Specific delivery and targeting of injured tissues could also be achieved through novel nanotechnology approaches. Recently, Karakus et al. formulated and characterized chemically modified polymeric nanoparticles (NPs) incorporating T3 in the surface in order to target membrane integrin receptor αvβ3 in such a manner that only the nongenomic effects are activated [78]. Modified T3 was conjugated to polylactide-co-glycolide (PLGA), which is a biodegradable and hydrophobic FDA-approved drug delivery carrier, in order to enhance T3 delivery and restrict its nuclear translocation [79,80]. Interestingly, PLGA-T3 NPs showed an enhanced cardioprotective effect, improved mitochondrial function, energy status, and preserved cytoskeletal integrity under hypoxic conditions. This nanotargeted delivery of T3 can prolong the circulation half-life of T3 and allows for the encapsulation of different bioactive molecules such as phosphocreatine, which could further improve the energy status of cardiomyocytes. In the field of HFpEF, nanomedicine-based approaches could be of utmost importance for the specific delivery of TH in the heart and the avoidance or minimization of TH-associated adverse effects. One recent attempt (funded by the EuroNanoMed III) incorporates computational, chemistry, and cellular biology approaches to develop a nanoparticle-based drug delivery system that will target and deliver T3 in diabetes-injured cells in order to restore cardiac and renal function. Both polymeric and lipid nanoparticles are functionalized with specific molecules in the surface that permit targeting and uptake from stressed cells. After uptake, T3 is released in the cell and acts on TRs. The main advantages of these smart nanocarriers are that adverse effects such as tachycardia, arrhythmias (mainly atrial fibrillation), kidney hyperfiltration, nervousness, and disruption of the thyroid axis could be avoided [81].

## 4. Conclusions

HF greatly affects patients’ quality of life and survival, and novel therapeutic approaches for its treatment are quite necessary. TH signaling, on the other hand, has a crucial role in the pathophysiology of HF and ischemic heart disease, while past and ongoing studies are promising, regarding TH’s therapeutic results. Novel or complementary therapeutic approaches with the use of cutting-edge technologies could become an important tool to highlight and enhance TH therapeutic potential, at the same time avoiding probable adverse effects.

## Figures and Tables

**Figure 1 biomedicines-11-00975-f001:**
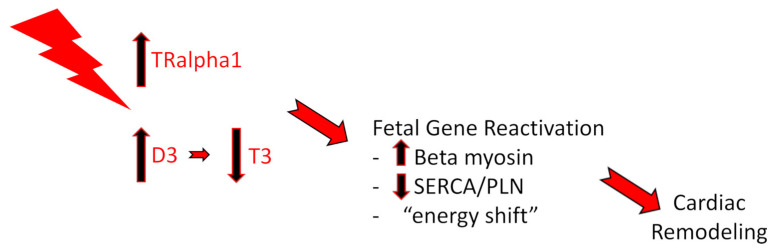
After stress, TRalpha1 increases in the nucleus while T3 is inactivated by D3. As a consequence, TRalpha1 acts as an aporeceptor, leading to fetal gene reactivation and cardiac remodeling.

**Table 1 biomedicines-11-00975-t001:** Clinical studies with TH administration in HF and AMI settings.

Clinical Study	Patients (N)	Setting	Treatment	Outcome	Safety
Hamilton et al. [46]	23	Advanced HF and low T3	0.15–2.7 ìg/kg (iv) T3 for 6–12 h	Increased CO and reduction in SVR	No AEs
Pingitore et al. [47]	20	HF and low T3	35.6 ìg LT3 (iv) in the first 24 h and 15 ìg/day until 72 h	Increased SV and lower HR; decrease in NT-proBNP, noradrenaline, and aldosterone	No AEs
Holmager et al. [48]	13	Stable systolic HF and low T3	20 μg oral T3 per day for 3 months	No changes in cardiac function and neurohormonal profile	No AEs
Amin et al. [49]	50	Chronic stable HF and low T3	T3 replacement dose by oral liothyronine for 6 weeks	Increased 6 min walk test, decreased hsCRP, decrease in NTproBNP	No AEs
Zhang et al. [50]	124 (estimated)	Chronic HF and low T3	Oral levothyroxine with a starting dose of 12.5 μg	Ongoing	Ongoing
Pingitore et al. [51]	37	AMI and low T3	Oral liothyronine (T3) (maximum dosage 15 mcg/m^2^/die for 6 months	Significant reduction in WMSI difference value (discharge/follow-up), increased stroke volume at follow-up	No AEs
Jabbar et al. [52]	95	AMI and subclinical hypothyroidism	Oral levothyroxine (25 µg titrated to serum thyrotropin levels between 0.4 and 2.5 mU/L	No significant differences	No AEs
Pantos et al. [56]	52	Anterior STEMI undergoing PCI	(i.v.) bolus injection of 0.8 μg/kg of LT3 followed by a constant infusion of 0.113 μg/kg/h i.v. for 48 h	Significantly lower LV end-diastolic volume index and LV end-systolic volume index at discharge, CMR IV tended to be lower in the LT3-treated group at 6 months	Tendency for an increased incidence of AF during the first 48 h

## Data Availability

Not applicable.

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
