# Peer review of "Thyroid Hormone and Heart Failure: Charting Known Pathways for Cardiac Repair/Regeneration"

_biomedicines, 2023, doi:10.3390/biomedicines11030975_

Round 1

Reviewer 1 Report

The paper “Thyroid hormone and heart failure: charting known pathways for cardiac repair/regeneration" by Mantzouratou et al deals with the assosiaciont between thyroid hormone and heart failure. 

Paper design is fine. The article is logically divided into sections and subsections. English is fine, only minor spell check needed. 

Comments: 

  1. A lot has been reported for hypothyroidism, but nothing has been reported about hyperthyroidism. Is there no evidence about it? From a clinical point of view, the concomitant presence of hyperthyroidism may induce tachycardia, thus increasing heart failure development and worsening due to cardiac increased consumption and reduced oxygenation. 
  2. Is there any evidence in heart transplanted patients? 

Author Response

We sincerely acknowledge your reviews and comments.

Thank you for your insight and contribution to this manuscript.

Replying to your comments

  1. Since this manuscript refers to the relation of thyroid hormone signaling and heart failure in terms of cardiac repair/regeneration (and not to thyroid disorders) we believe that an analysis of the undeniable effects of hyperthyroidism in the heart would not be of added value to the reader.
  1. It is true that there is evidence about transplanted patients and we have added a relevant paragraph (lines 324 to 329).

Reviewer 2 Report

Thank you for the invitation to review this interesting text.

The content fully corresponds to the topic, article is well-written and gives a clear picture of the actual state of knowledge. The literature in this area was thoroughly reviewed.

However, I have only one small suggestion: please note the notation of keywords: ‘heart failure’ - in the abstract the authors use capital letters, in the introduction the correct notation. Likewise ‘Thyroid Hormone’, Ejection Fraction’, ‘Myocardial Infraction’ - in the middle of a sentence, these names should be written in lowercase letters.

I believe that the article will attract a wide range of readers

Author Response

We sincerely acknowledge your comments and we thank you for your insight and contribution to this manuscript. We have corrected the notation of keywords as you correctly suggested.